# *Bifidobacterium*: Host–Microbiome Interaction and Mechanism of Action in Preventing Common Gut-Microbiota-Associated Complications in Preterm Infants: A Narrative Review

**DOI:** 10.3390/nu15030709

**Published:** 2023-01-30

**Authors:** Fatemah Sadeghpour Heravi, Honghua Hu

**Affiliations:** 1Macquarie Medical School, Macquarie University, Sydney, NSW 2109, Australia; 2Innovation Center of Translational Pharmacy, Jinhua Institute of Zhejiang University, Jinhua 321016, China

**Keywords:** gut microbiota, *Bifidobacterium*, probiotic, preterm infants, host-microbiome interaction

## Abstract

The development and health of infants are intertwined with the protective and regulatory functions of different microorganisms in the gut known as the gut microbiota. Preterm infants born with an imbalanced gut microbiota are at substantial risk of several diseases including inflammatory intestinal diseases, necrotizing enterocolitis, late-onset sepsis, neurodevelopmental disorders, and allergies which can potentially persist throughout adulthood. In this review, we have evaluated the role of *Bifidobacterium* as commonly used probiotics in the development of gut microbiota and prevention of common diseases in preterm infants which is not fully understood yet. The application of *Bifidobacterium* as a therapeutical approach in the re-programming of the gut microbiota in preterm infants, the mechanisms of host-microbiome interaction, and the mechanism of action of this bacterium have also been investigated, aiming to provide new insights and opportunities in microbiome-targeted interventions in personalized medicine.

## 1. Introduction

The gastrointestinal tract, which houses trillions of microorganisms, is the most populated anatomical niche in the human body and plays a critical role in the development of the immune system, metabolism, cognitive development, and host physiology [1]. 

The gut microbiota structure is constantly changing during life in infancy and childhood and stabilizing through adulthood [2]. Different prenatal and postnatal factors can influence the structure and composition of the gut microbiota including delivery method, genetics, feeding method, maternal microbiota, antibiotics, and lifestyle.

Dysbiosis, or the disruption of the gut microbiota, has been associated with the development of a number of chronic illnesses in premature newborns, which may persist later in adulthood, including gastrointestinal disorders, neurodevelopmental and metabolic abnormalities, and allergies [3]. 

Although preterm infants’ health outcomes are equally relevant and important, the majority of research on gut microbiota has focused on full-term infants and adults. According to the World Health Organization (WHO), 15 million infants are delivered prematurely each year. Complications associated with prematurity are the major reason for 1 million deaths among children under 5 years of age each year, survivors may also face lifetime mental and physical challenges [4].

Preterm newborns are immunologically underdeveloped, making them vulnerable to bacterial infections. Neutropenia, deficiency of phagocytosis, chemotaxis, the cytolytic activity of NK cells, low expression of histocompatibility complex class II, and suppressed toll-like receptor (TLR) are the most common immunodeficiencies in preterm infants [5,6]. Preterm infants born before 37 weeks of gestational age (weight < 2500 g) may be exposed to different environmental factors including long-term stays in the neonatal intensive care units, use of broad-spectrum antibiotics, and monitored feeding regimens [7]. Although maternal milk contains several beneficial components such as antimicrobial peptides, immunoglobulins, essential nutrients such as proteins, Zinc, lactoferrin, natural probiotics, Fructooligosaccharides (FOS), short-chain galactooligosaccharides (GOS), and polydextrose, not all preterm infants can digest their mother’s milk and absorb its nutritional substances [8]. Therefore, preterm infants with underlying health conditions require additional nutritional support to maintain gastrointestinal health and absorption of essential nutrients [9].

The classic pattern of the gut microbiota in a full-term, vaginally born, and breastfed infant follows a general trend that includes initial colonization with facultative anaerobes including *Enterobacteriaceae* family (e.g., *Escherichia* coli, *Klebsiella* spp.), *Enterococcus* spp., *Streptococcus* spp., and *Staphylococcus* spp. After depletion of oxygen by facultative anaerobes in a matter of days after birth and diet shift to human milk, which is a rich source of oligosaccharides, obligate anaerobes and oligosaccharides metabolizers such as *Bifidobacterium* spp., *Bacteroides* spp., and *Clostridium* spp. dominate the gut [10]. Subsequently, solid food consumption by infants after the age of six months reduces *Bifidobacterium* abundance by 30% to 40%, and this decline persists throughout childhood and adolescence as a result of lifestyle, puberty, nutrition, and antibiotic administration [11]. In adulthood, *Bifidobacterium* abundance stabilizes between 0% to 18% and declines in elderlies which might be related to declined immune function in this group [12].

Recent investigations using culture-based and sequencing-based approaches have found a strong association between the function of *Bifidobacterium* in the development of inflammatory intestinal diseases, neurodevelopmental disorders, and allergies in premature infants [13]. In addition to the numerous correlations observed, a substantial body of evidence has shown the beneficial impact of *Bifidobacterium* in a range of preclinical and clinical models. However, it remains unclear how this interaction can lead to the regulation of immunological pathways and the improvement of the immature gastrointestinal tract. 

To gain a mechanistic understanding of host-microbiome interaction and how bacterial metabolites can remotely regulate other organs and pathways, we discussed the impact of *Bifidobacterium* on host metabolism and physiology in pre-term infants, aiming to provide new insights and opportunities in microbiome-targeted interventions in personalized medicine in this population. 

## 2. Common Gut-Microbiota-Associated Complications in Preterm Infants

### 2.1. Gastrointestinal Disorders

#### 2.1.1. Necrotizing Enterocolitis (NEC)

Necrotizing enterocolitis (NEC) is the most common intestinal complication in preterm infants. NEC is a devastating condition defined as intestinal inflammation/perforation (ischemic necrosis of intestinal mucosa) that mainly occurs during the first two weeks of life in 10% of preterm infants. Preterm infants diagnosed with NEC may experience severe symptoms including lethargy, bloated stomach, vomiting, blood in stool, multiorgan failures such as slow heartbeat (bradycardia), difficulties in breathing (apnea), and even death [14] (Figure 1). According to a systematic and meta-analysis review of 574,692 premature infants, the global incidence of NEC was predicted in seven out of 100 preterm infants [15]. 

Despite many research efforts on the management of NEC over the last decades, NEC risk in preterm infants still is high and survivors may experience long-term consequences. Current management of NEC includes a controlled diet through a nasogastric tube, administration of inotropes and intravenous fluids to maintain oxygen delivery to different organs, and prevention of enteric bacterial infection using broad-spectrum antibiotics. Severe cases may require abdominal surgery to resect the necrotic tissue and drainage of fluid from the peritoneal cavity [16]. 

Recently, gut microbiota dysbiosis has been identified as one of the main factors in the development of NEC in preterm infants. Several studies have shown the association of NEC incidence with a high abundance of Gram-negative facultative bacteria (e.g, *Proteobacteria* and *Gammaproteobacteria* (*Enterobacteriaceae* members (*Klebsiella pneumoniae*, *E. coli*, and *Enterobacter cloacae*), *clostridia* (*C. neonatale, C. butyricum,* and *C. perfringens*)), and a low abundance of obligate anaerobic bacteria such as *Bifidobacterium* (*B. longum* sp. *Infantis*), *Bacteroides* spp., and *Clostridium* spp. [13,17,18].

Uncertainty surrounds how dysbiosis in gut microbiota affects NEC pathogenesis, however, results of piglet, mice, and human studies suggest that stimulation of immature enterocytes by Gram-negative lipopolysaccharide through Toll-like receptor 4 (TLR4) can lead to over-activation of inflammatory responses in the intestines of premature infants and lead to bowel damage and NEC progression [19,20]. In a study conducted by Cynthia et al. [21], TLR4- deficient *C3H/HeJ* mice did not develop NEC, whereas wild-type *C3H/HeOUJ* genotypes had a significant chance of developing NEC. This may imply the impact of TLR4 over-expression in mucosal damage, death of enterocyte cells, and bacterial translocation into bodily fluids [22].

Other studies have also shown how TLR4 prevention factors such as nucleotide-binding oligomerization domain-containing 2 (NOD2) receptor (CARD15) could prevent NEC onset. TLR4-NOD2 inhibitory interaction in enterocytes protected intestinal mucosal from NEC development. In this study enterocytes without TLR4 or NOD2 were assessed in intestinal-specific wild-type mice or mice with intestinal-specific wild-type or dominant-negative TLR4 or NOD2, and in mice with NEC. The result showed that NOD2 could prevent TLR4 expression and enterocyte apoptosis in mice models [23]. Another study has also shown the impact of Recombination-activating gene 1 (RAG1) deficiency (an essential gene in T and B lymphocyte development) in the onset of NEC. In this study, (Rag1−/−) deficient mice were protected from NEC while transferring intestinal lymphocytes from NEC mice into naive mice triggered intestinal inflammation. Moreover, inhibition of IL-17 or STAT3 (an essential factor in the differentiation of TH17 helper) lowered the risk of enterocyte proliferation and NEC in this study [24]. Gram-negative bacteria such as *Enterobacteriaceae* members can also influence the activation of TLR4 in the enterocyte. Preterm infants with NEC have an overabundance of LPS-producing bacteria, which could lead to the over-stimulation of TLR4. LPS-enriched gut microbiota (particularly *Enterobacteriaceae*-dominated microbiota) has been associated with a higher risk of epithelial necrosis and NEC in preterm infants, while bacterial communities with lower CpG DNA (potent activator of TLR4 and TLR9) have been associated with a lower risk of NEC [25]. Other studies have also shown the association of NEC with *Enterobacteriaceae* dominance. In Greenwood et al.’s study on 74 preterm infants with and without antibiotic administration, preterm infants who received antibiotics showed a different microbial pattern compared to the control group. Early antibiotic exposure led to a higher abundance of *Enterobacter* in preterm infants which may be associated with the over-activation of TLR4 and a higher risk of NEC incidence [26].

#### 2.1.2. Late-Onset Sepsis (LOS)

Sepsis is a medical emergency that requires early diagnosis and treatment in neonates. Sepsis defines as a blood infection by pathogenic microorganisms. According to a large neonatal population-based meta-analysis study from 12 middle-income and high-income countries on four continents, the number of neonatal sepsis for each 100,000 live births was estimated at 2202 cases with a mortality rate of 11–19% or 3.0 million cases annually [27]. Neonatal sepsis may occur during the first 72 h of life by mother-to-infant pathogen transmission before or during delivery (early-onset sepsis) or it can develop later in life through hospital-associated pathogen transmission or the translocation of pathogens from the gut to the bloodstream (late-onset sepsis) [28]. The current management approach of LOS is limited to antimicrobial therapy and adjunctive therapy by increasing neutrophil quantity (e.g., granulocyte-macrophage colony-stimulating factor (GM-CSF), granulocyte transfusions, and intravenous immune globulin (IVIG) [29]. Although no specific bacterial taxa have been detected as the causative agent of LOS, recent studies have shown the association of various bacterial species to the onset of LOS. The development of LOS has been linked to a low *Bifidobacterium* abundance and a high abundance of Gram-negative bacteria such as enteric bacteria (*E. coli*, *Pseudomonas* spp., and *Klebsiella* spp.), coagulative-negative *Staphylococci* (*CoNS*), and Gram-positive bacteria (*Enterococcus* spp. and *Streptococcus* spp.) [30,31].

### 2.2. Allergies

Inadequate early exposure to immune system modulator factors during the crucial newborn period may result in low immunological tolerance and an exaggerated immune response to endogenous and exogenous antigens and lead to the development of allergic diseases in preterm infants [32].

Atopic disease is a broad phrase for explaining various allergic diseases in children and atopy is the overactivation of the IgE-mediated immune response to allergens, which causes a variety of allergic disorders, including food allergy, asthma, atopic dermatitis, and rhinitis [33]. Pro-allergic pathways, which are activated as the result of imbalanced Th1, Th2, and Treg phenotypes, increased secretion of IL-4, IL-5, IL-13, and low secretion of IFN-γ by Th2, can lead to the development of different allergic disorders. Activation of pro-allergic pathways can be controlled by the gut microbiome, which maintains the Th1-Th2 balance and regulates Th17 and Treg cells [34].

In healthy conditions, mature Th1 and Treg can regulate the Th2 phenotype and prevent the activation of proinflammatory cytokines [35]. Even though dysbiosis can lead to the development of allergic diseases in preterm infants, the supplementation of different *Bifidobacterium* strains in the regulation of anti-allergic pathways has shown promising results in the prevention of allergies. 

Studies applying sequencing-based approaches have shown the association of allergic diseases with lower gut microbial diversity and lower abundance of *Bifidobacterium* strains between non-allergic and allergic infants [36,37,38,39,40]. For instance, Guo et al.’s study has shown that infants with cow’s milk protein allergy had lower *Bifidobacterium* diversity, which may explain the key role of *Bifidobacterium* in the digestion of essential components in milk and the gut-immune system crosstalk in infants. Moreover, a case-controlled investigation on 21 toddlers revealed different gut bacterial compositions between children with and without atopic dermatitis (AD). This study demonstrated the considerable long-term effects of immature gut microbiota on the development of allergies even after infancy by demonstrating significantly decreased *Bifidobacterium* abundance in children aged 3 to 5 with eczema [41]. The association of allergic diseases such as atopy and asthma with a low abundance of *Bifidobacterium*, *Faecalibacterium*, *Akkermansia*, and *Faecalibacterium* was also reported in a follow-up study on 308 children aged 1–11 months [42].

Abrahamsson et al. investigated the microbial diversity of 47 infants during the first year of life and school-age at 7 years old. This study showed that lower bacterial diversity was associated with an increased risk of subsequent allergic disease, while bacterial phyla/genera abundance did not differ significantly in children with and without allergic diseases [43]. The author has also previously shown the association of IgE-associated eczema with low gut microbial diversity in the same study population [44]. In a larger sample size, a meta-analysis study on 147,252 children showed that preterm infants with younger gestational age were at a high risk of preschool wheezing and school-age asthma. The risk of allergic diseases such as food allergies was also investigated on 13,980 preterm infants [45]. However, this study did not report any significant statistical difference in the risk of food allergy with prematurity.

### 2.3. Neurodevelopmental Diseases

Gut-brain axis is shaped during prenatal and postnatal life, therefore, imbalanced gut microbiota can have a significant effect on the nervous system and brain development [46]. Imbalanced gut microbiota can impact different domains of cognitive trajectories such as learning and memory, complex attention, social cognition, and executive function [47]. Among different microbial metabolites, short-chain fatty acids seem to be the main mediators in the gut-brain crosstalk [48]. However, reciprocal interaction and pathways involved in this crosstalk have not been fully understood yet. 

#### 2.3.1. Attention Deficit Hyperactivity Disorder

Attention deficit hyperactivity disorder (ADHD) is largely a heritable mental disorder, however, recent findings have shown the association of environmental factors such as nutrition and gut microbiota on the onset of ADHD. Studies have shown that pro-inflammatory inducer molecules of gut microbiota such as TNF, IL-6, and IL-1β could stimulate the brain’s innate immune system and lead to neuroinflammation and neurodevelopmental abnormalities [49]. Mouse transformation models with preterm infants’ gut microbiota induced systematic pro-inflammatory mediators such as TNF, IL-1β, IFNγ, and NOS1 in the brain, which emphasized the impact of gut microbial structure and its metabolites on neuroinflammation and brain development [50]. Furthermore, studies on adults diagnosed with attention deficit hyperactivity disorder (ADHD) have shown a different gut microbial composition in the ADHD population compared to healthy individuals. For example, Aarts et al. showed that ADHD cases had an increased abundance of *Actinobacteria* genus (particularly *Bifidobacterium*; controls: 12.66% to ADHD: 20.47%; *p* = 0.002). Nevertheless, this study did not investigate the functional effect of *Bifidobacterium* metabolites on the onset of neurodevelopmental disorders, which should be taken into account in the management of neurological disorders using gut microbial signature [51].

#### 2.3.2. Schizophrenia Spectrum Disorder

Prospective research on neurodevelopmental outcomes in preterm infants has shown that premature infants are at a higher risk of psychotic disorders such as schizophrenia. In addition, they have a 2.9 times higher risk of developing serious depression and 7 times higher risk of bipolar illness, and a 3.5 times greater chance of developing eating disorders in their childhood and adulthood [52]. According to the Nosarti et al., study, infants born prematurely are at higher risk of hospitalization due to different psychiatric disorders [53]. Although limited studies are available on the investigation of gut microbiota with schizophrenia in preterm infants, it has been shown that patients with the first episode of psychosis showed a higher abundance of *Lactobacillus*, *Bifidobacterium*, and *Ascomycota* [54,55]. 

#### 2.3.3. Autism Spectrum Disorder

Poor social communication skills and restricted patterns of repetitive behavior known as autism spectrum disorders (ASD) are other adverse neurodevelopmental outcomes that may develop in preterm infants. Preterm infants have a 3.3 times higher chance of autism diagnosis than full-term infants [56]. Clinical studies have shown an imbalanced gut microbial composition and metabolites in preterm infants with ASD. However, there are discrepancies in the findings which may be related to the antibiotics administration as well as different study designs and methodologies. 

A systematic review conducted on 15 cross-sectional studies showed incompatible findings on gut microbial composition between ASD and non-ASD populations. Based on this study three major phyla; *Firmicutes*, *Bacteroidetes*, and *Proteobacteria* showed the highest variations between ASD and non-ASD populations. This study has shown a lower abundance of *Bifidobacterium* in the ASD group [56]. Recent metabolomics studies have also shown higher concentrations of short-chain fatty acids and lower concentrations of phenylacetylglutamine, hippurate, and 4-cresol sulfate in the ASD group compared to non-ASD controls [57].

## 3. General Characteristics of *Bifidobacterium*

Members of the *Bifidobacterium* genus are the most prevalent bacterial community forming 40 to 90% of the total gut microbiota at different developmental ages. *Bifidobacteria* are gram-positive, non-spore-forming anaerobic bacteria with pleomorphic rod morphology [58]. *Bifidobacterium* was first isolated from fecal samples in healthy breastfed infants by Henri Tissier at the Pasteur Institute in France in 1899 [59]. *Bifidobacterium* belonging to the *Actinobacteria* phylum has 94 recognized (sub) species classified in seven clusters including *Bifidobacterium longum*, *Bifidobacterium bifidum*, *Bifidobacterium adolescentis*, *Bifidobacterium boum*, *Bifidobacterium pullorum*, *Bifidobacterium asteroids*, and *Bifidobacterium pseudolongum* [60,61]. *Bifidobacterium longum* (subsp. *Infantis*), *Bifidobacterium breve*, and *Bifidobacterium bifidum* are common colonizers in the early stages of life, while *Bifidobacterium adolescentis* are associated with adulthood [62,63]. 

Successful adaptation of *Bifidobacterium* to the human gastrointestinal tract from infancy to adulthood may be explained by the presence of many genes attributed to stomach acid tolerance, metabolism of carbohydrates, and transport systems in the *Bifidobacterium* genome [64]. The average genome size of *Bifidobacterium* is 2.44 Mb with an average of 58.91% G + C content containing a large number of genes involved in the complex metabolism of human milk oligosaccharides (HMOs) [61]. Fermentation of HMOs by *Bifidobacteria* using glycosyl hydrolases produces short-chain fatty acids (SCFAs), which have many health-promoting properties including maintenance of intestinal barrier integrity and anti-inflammatory functions [65]. Moreover, the metabolism of aromatic amino acids (phenylalanine, tryptophan, and tyrosine) by *Bifidobacterium* produces aromatic lactic acids (4-hydroxyphenyl acetic acid (4-OH-PLA), indolelactic acid (ILA), and phenyllactic acid (PLA)), which have anti-inflammatory and antibacterial activities [66]. 

Even though recent studies have shown promising results in the administration of *Bifidobacterium* as a probiotic in the development of the gut microbiota in preterm infants, it is still unclear how *Bifidobacterium* abundance and its metabolites are inversely associated with the development of several life-threatening diseases in prematurely born infants.

## 4. Immunomodulatory Effects of *Bifidobacterium*

The gut-associated lymphoid tissue (GALT) is the largest mass of lymphoid tissue in the human body and contains a variety of immune cells, including B and T lymphocytes, as well as antigen-presenting cells such as dendritic cells (DC) and macrophages. 

Intestinal epithelial cells provide a protective layer between intestinal mucosa and luminal microorganisms (Figure 2). For instance, Goblet and Paneth cells secrete mucus layer and antimicrobial peptides, respectively, to enhance protective effects against luminal microorganisms in the gastrointestinal tract. Secretory immunoglobulin A (sIgA) secreted by B cells have also protective roles against luminal microbiota [67]. 

Gut microbiota-immunity crosstalk can activate different immunological pathways either in a regulated or exaggerated way and lead to the development of several diseases including gastrointestinal and dermatological disorders, allergies, and host behavioral changes [68]. This interaction is activated by beneficial microbiota and pathogens through different recognition receptors which are highly expressed in intestinal epithelial cells (IECs) including pattern-recognition receptors (PRRs), Nucleotide-binding and oligomerization domain (NOD)-like receptors (NLRs), Toll-like receptors (TLRs), C-type lectin receptors (CLRs), RIG-I-like receptor (RLR), Absent in melanoma 2 (AIM2)-like receptors (ALRs), and the oligoadenylate synthase receptor (OAS) [69]. Activation of PRRs can lead to the production of different antimicrobial peptides (AMPs) such as α- defensins (HD5, HD6) and regenerating islet-derived protein III (REGIII α, β, and γ) by immune cells and intestinal Paneth cells and restrict the access of pathogens to the mucosal epithelium [70]. Host-microbe interaction can also influence T cell differentiation into Th1, Th2, Th17, and regulatory (Treg) cells, which are regulated by pro-inflammatory and anti-inflammatory cytokines such as transforming growth factor-β (TGF β) and interleukin-10 (IL-10) [71]. 

The antibacterial and antiviral effects of *Bifidobacterium* against various pathogenic microorganisms have been the subject of numerous studies. For instance, using human colorectal adenocarcinoma cell lines (HT-29), *B. longum* has been proven to have an inhibitory effect against Gram-negative bacteria including *Salmonella typhi STN12, Salmonella enteritidis SEN6, Escherichia coli EC4219,* and *Escherichia coli EC3960*. Although this investigation has primarily focused on the prevention effect of *B. longum* on adhesion activities of Gram-negative pathogens, under in vivo conditions, a variety of contributing factors, such as intestinal epithelial cells (IECs) and IECs’ tight junction can determine how *Bifidobacterium* acts antagonistically [72]. Other studies have also shown the inhibitory effect of *Bifidobacterium* strains such as *B. longum, B. adolescentis*, and *B. pseudocatenulatum* against multidrug-resistant pathogens (e.g., *E. coli*), Vancomycin-resistant bacteria (*Enterococcus* and *Staphylococcus aureus*) using in vitro human cell line models and animal models [73,74,75]. 

*Bifidobacteria* has also been proven in numerous studies to have antiviral effects in mice models and colonic cells [76,77,78,79,80]. For instance, in Caco-2 and HT-29 cells, *B. thermophilum RBL67* showed anti-rotaviral activities. According to this investigation, *B. thermophilum RBL67* had greater adhesion indices on Caco-2 and HT-29 cells than *B. thermacidophilum* isolated from newborn fecal samples (*RBL69* and *RBL70*). However, to confirm the inhibitory effects of *Bifidobacterium* strains on bacterial and viral infections further studies in human-like models are needed [76]. 

*Bifidobacterium* strains can also contribute to the regulation of pro-inflammatory and anti-inflammatory cytokines; in a case-control study, the intervention population who consumed dairy products containing *B. lactis* and other beneficial strains showed higher serum levels of pro-inflammatory cytokines (interferon-γ (IFN-γ), interleukin 12 ((IL12), and immunoglobulin (Ig)) and higher activity in natural killer cells, which may suggest the effectiveness of the *Bifidobacterium* in the improvement of immune responses and NK cell functions [81]. *Bifidobacterium* strains can also induce macrophage mediators and modulate host immune responses. It was also shown that *B. pseudocatenulatum SPM1204* isolated from fecal samples cultured with dendritic cells and macrophages increased histocompatibility complex (MHC) class I and induced the production of nitric oxide (NO), tumor necrosis factor (TNF), and IL1 [82]. Table 1 shows the *Bifidobacterium* role as a probiotic in preterm infants in human studies.

## 5. *Bifidobacterium* as Probiotic

According to the definition introduced by the Food and Agriculture Organization of the United Nations and the World Health Organization (FAO/WHO), probiotic is live microorganisms that when administered in adequate amounts confer a health benefit on the host [118].

Nutrition has been identified as the main approach in the regulation of host physiology and programming of the gut microbiota in the early stages of life. The significant impact of environmental factors including diet on gut microbiota brought the new concept of “Early-life Nutritional Programming” theory during the first three years of life which has long-lasting consequences throughout the lifespan [119]. Reprogramming of gut microbiota by maintaining the balance of beneficial bacterial species through the administration of probiotic strains can prevent several microbiota-associated infections in preterm infants and protect survivors from severe morbidity. 

Owing to the protective and immunomodulatory effects of *Bifidobacterium* in early life, the European Food Safety Authority (EFSA) has approved the Qualified Presumption of Safety (QPS) status of different species of *Bifidobacterium* including *B. longum, B. breve*, *B. bifidum*, *B. adolescentis*, and *B. animalis* [120].

Human milk is the first source of *Bifidobacterium* species and predominates in breastfed infants during the first three years of life. The metabolism of human milk oligosaccharides by *Bifidobacterium* species can alter gut microbial composition and promote immune system development. *B. longum* subsp. *Infantis, B. longum* subsp. *Longum*, *B. bifidum*, and *B. breve* are the most commonly identified species in newborn infants [121]. The primary protective function of *Bifidobacterium* species is their overabundance in human milk, which can result in higher *Bifidobacterium* colonization in the gastrointestinal tract, particularly in the colon, as the most suitable niche for this bacterial community [122]. Additionally, in vivo investigations have demonstrated that *Bifidobacterium* species grown on HMO have excellent adhesion abilities to intestinal epithelial cells, which is essential to compete with opportunistic pathogens [123]. Additionally, prebiotics such as HMO and lactoferrin in human milk which are human non-digestible beneficial components can be metabolized by the gut microbiota and promote the growth of beneficial microorganisms and prevent the overgrowth of pathogenic microorganisms in the gastrointestinal tract [124]. 

### 5.1. Bifidobacterium and Prevention of NEC and LOS

Recent findings suggested probiotics as the most effective human intervention in the management of LOS and NEC. According to a systematic analysis of 44 observational, randomized controlled, and RCTs studies, probiotics could reduce the sepsis rate by 12% in RCTs and 19% in observational studies in preterm infants. This study has also shown a slight reduction in NEC incidence in observational studies. The results suggested the beneficial effect of probiotics in the prevention of late-onset sepsis, NEC, and mortality rate in preterm infants [125]. Another meta-analysis review of 16 studies including 2842 preterm infants revealed a significant impact of probiotic supplementation on NEC incidence (typical RR 0.35, 95% CI 0.24 to 0.52) and mortality rate (typical RR 0.40, 95% CI 0.27 to 0.60), while no significant reduction was reported on sepsis incidence (typical RR 0.90, 95% CI 0.76 to 1.07) [126]. The effectiveness of specific probiotic strains in NEC prevention was also evaluated in a meta-analysis review of 26 studies. In this study findings from 6605 infants (placebo: 3281 and probiotic: 3324) showed that the relative risk of NEC was significantly lower in infants receiving probiotics compared to the placebo group (0.47 (95% CI 0.36–0.60) *p* < 0.00001). Studies using *Lactobacillus GG* [127,128], *Lactobacillus reuteri* [129,130], *Lactobacillus sporogenes*, and *Saccharomyces boulardii* [131,132,133] showed no significant reduction in NEC incidence (0.62 (95% CI 0.37–1.05), *p* = 0.07) [134]. In contrast, studies using *B. lactis* [86,87,92,101], *B. breve* [83,97], *B. bifidum* [135] showed a significant reduction in relative risk of NEC in the probiotic group (0.24 (95% CI 0.10–0.54), *p* = 0.0006). 

Investigation of *Bifidobacteria’s* role in the prevention of gastrointestinal disorders in animal models has also shown encouraging results. The prevention role of *Bifidobacteria* on intestinal microbes’ invasion from mucosa to internal organs showed that bacterial translocation in Peyer’s patches in mice models decreased by higher *Bifidobacteria* colonization in caecum and colon and prevented blood, liver, and lungs infections, while colonization of other pathogenic microorganisms such as *Bacteroides fragilis* and *clostridia* were associated with increased risk of bacteremia and lung infection in these models [136]. Also, the transcriptional activity of enterocytes and regulation of innate immune-mediated inflammation in mice models has shown that administration of *B. infantis* downregulated the expression of IL8, IL6, TNFα, IL23, iNOS, and antimicrobial peptides and altered the expression of intestinal mucus-related proteins and led to the low incidence of NEC in animal models [137]. *B. infantis* administration has also been associated with enhanced expression of tight junction proteins (4 Claudin and occludin), and a low incidence of NEC in the neonatal mouse NEC model [138].

Some studies have also shown the higher effectiveness of multiple species compared to single-species probiotics. For instance, the comparison of daily administration of a single strain of *B. breve M-16V* (5 × 10^8^; one-species group) and a combination of three species *B. breve* M-16V, *B. longum* subsp. *infantis M-63* and *B. longum* subsp. *longum BB536* (5 × 10^8^ of each strain; three-species group) for one week has shown that *Bifidobacterial* fecal count was significantly higher in preterm infants who received three-species probiotics compared to the one-species group. Moreover, the abundance of pathogenic bacterial species such as *Clostridium* and *Enterobacteriaceae* was significantly lower in preterm infants who received three-species probiotics [95]. Combination of probiotic strains including *B. longum* subsp. *infantis BB-02, B. animalis* subsp. *lactis BB-12,* and *S. thermophilus TH-4* in 459 preterm infants (probiotic: 229 and placebo: 230) could also increase the abundance of probiotic species in the gut microbiota of preterm infants which may imply the importance of early administration of multi-strain probiotics on the abundance of beneficial bacterial species in preterm infants [139]. Likewise, a comparison of 119 preterm infants who received human milk with probiotics (combined supplementation of *B. breve* and *Lactobacillus casei*) with 112 preterm infants receiving human milk without probiotics showed that supplementation *of B. breve* and *L. casei* reduced the NEC occurrence [93]. 

Metabolomic studies have also shown the association of probiotic supplementation with variation in concentration of beneficial health indicators such as short-chain fatty acids (SCFAs) (acetate and lactate) in preterm infants. Short-chain fatty acids are one of the primary microbial byproducts of the breakdown of human milk oligosaccharides and indigestible fiber [140]. Primary colonization of gut microbiota with lactate-producing bacteria (e.g., *Bifidobacterium*, *Lactobacillus*, and *Bacteroides*) in infants has beneficial effects on the maturation of epithelial cells and mucosal dendritic cells. As a result, the level of fecal SCFAs can indicate microbial structure and state of health in infants. According to an observational longitudinal study on 234 preterm infants (probiotic:101 and placebo:133), supplementation of *Bifidobacterium* and *Lactobacillus* was associated with higher fecal acetate and lactate and a lower fecal pH in the probiotic group compared to the placebo group. A higher concentration of acetate and lactate may show the exceptional ability of the *Bifidobacterium* strain in metabolizing human milk oligosaccharides into SCFAs [110]. Another study investigating the role of *B. lactis Bb12* supplementation on health indicators of preterm infants showed that in preterm infants receiving probiotic fecal pH and calprotectin (an indicator of gastrointestinal disorders) were significantly lower compared to the placebo group, while fecal concentrations of acetate, lactate, and IgA were significantly higher in the probiotic group compared to the placebo group [86]. 

While some research found encouraging results in the prevention of common complications in preterm infants using probiotic strains, other studies have shown no association in this regard. For instance, a single-center retrospective study of 293 preterm infants (37 NEC cases) who were routinely supplemented with a multispecies probiotic for 4 years prior to and 5 years after probiotic administration (n = 14, n = 23, respectively) showed no significant difference in NEC rate [115]. However, these findings may show an already low rate of NEC rate in this center, and a multi-center retrospective analysis is needed to determine the beneficial effects of probiotics in NEC reduction and mortality. Similarly, a randomized controlled study on 1315 preterm infants (probiotics:650 and placebos: 660) showed that preterm infants who received daily *B. breve BBG-001* over 6 weeks showed no significant reduction in NEC rate and late-onset sepsis compared to the placebo [99]. Routine administration of *B. breve M-16V* (1 mL = 1.5 billion CFU) in preterm infants and full-term infants also did not reduce the NEC and LOS rate between preterm (n = 162) and full-term infants (n = 1218) in a similar study [114]. Underestimation of the beneficial effects of probiotic strains may also be caused by cross-contamination of the placebo and probiotic participants or unsuccessful colonization of probiotic strains in the probiotic group due to antibiotic use or gastrointestinal immaturity. However, ignoring the effectiveness of probiotic strains might be a simple conclusion; larger randomized controlled trials are needed to evaluate the impact of probiotics, prebiotics, or a combination of both known as symbiotics on the prevention of common complications in preterm infants. 

### 5.2. Bifidobacterium and Prevention of Neurodevelopmental Diseases

Due to the anti-inflammatory effects of probiotics including the prevention of brain tissue infection such as white matter infection and modulation of brain development through regulation of immune cytokines, hormones, and neurotransmitters, probiotics may have neuroprotective effects in preterm infants. Recent findings suggest that early exposure to probiotics in preterm infants may be protective against neurodevelopmental disorders such as attention deficit hyperactivity disorder (ADHD) and autism spectrum disorder (ASD) [141]. For instance, Partty et al., have investigated the association of early probiotic intervention with neuropsychiatric disorders. In this study, 75 eligible infants (probiotic:40 and placebo:35) have been recruited. This study has shown that infants who received probiotics (*Lactobacillus rhamnosus GG* (ATCC 53103)) during the first 6 months of life which were followed up for 13 years had a lower rate of ADHD disorder compared to the placebo group 6/35 (17.1%). In contrast, none of the infants receiving probiotics was diagnosed with ADHD (*p* = 0.008). *Bifidobacterium* abundance was also lower in children diagnosed with ADHD during their infancy than in children without any neurodevelopmental disorder [142]. Another long-term follow-up study on 67 preterm infants (probiotic:36 and placebo:31) has also found that supplementation of *B. breve M-16V* (commonly isolated from human milk) did not have any significant effect on different developmental skills (e.g., language, learning, and memory, executive ability ad attention, social skills, sensorimotor functioning, and visuospatial processing) at 3 to 5 years age in preterm infants [111]. Also, combined probiotic treatment using *B. infantis, B. lactis, and Streptococcus thermophilus* on 1099 very preterm infants (probiotic:548, and placebo:551) over 2 to 5 years, showed no adverse neurodevelopment and behavior changes later in childhood. In this study, the development of infants was assessed across cognitive, language, and motor development domains following the Bayley-III tool [106]. Although some of these studies have been limited to a low number of participants and a low follow-up rate, the findings may be useful in designing long-term follow-up studies on the safety and long-term effects of probiotic administration in preterm infants.

## 6. *Bifidobacterium*: Mechanism of Action

Probiotic strains can modulate the host immune system through several mechanisms (Figure 3). Major mechanisms of action include modulation of adaptive and innate immunity, enhancement of intestinal epithelial barrier, prevention of pathogen adhesion, and production of antimicrobial compounds, which have been discussed in detail as follows.

### 6.1. Modulation of the Immune System

The innate immune system also known as the nonspecific immune system is the first line of defense in the human body including the protective effects of skin and mucosal membrane and immune system cells. While the adaptive immune system is a specific immunity to identifying pathogens by specialized immune cells including B and T lymphocyte cells [143]. Probiotics can modulate innate and adaptive immunity and lead to the enhancement of intestinal epithelium through immune mediators such as Toll-like receptors (TLRs), cytosolic signaling receptors such as nucleotide-binding oligomerization domain leucine-rich repeat-containing and pyrin domain-containing (NLRP), and anti-inflammatory cytokines.

### 6.2. Intracellular Immune Receptors (TLRs, NLRs) and Anti-Inflammatory Mediators

Intracellular immune receptors have a remarkable role in recognizing pathogen-associated molecular patterns (PAMPs) and microbial signals. Toll-like receptors (TLRs) are highly expressed in immune cells (dendritic cells, macrophages, and Natural killer cells (NK)) and non-immune cells (endothelial and epithelial cells). Recognition of microbial compounds by TLRs leads to the activation of downstream immune responses and the production of several inflammatory cytokines and other immune mediators which lead to innate and adaptive immune responses [144]. Enterocytes or intestinal absorptive cells line the inner surface of the intestine and express TLR4 as abundant proteins on their outer surface which are in close contact with microbial compounds in the gut lumen. TLR4 can recognize Lipopolysaccharide (LPS) in Gram-negative bacteria and activate MYD88 protein (myeloid differentiation primary response 88). Activation of MYD88 leads to kinase activation and degradation of NFκB/IKB dimer (Nuclear factor kappa-light-chain-enhancer of activated B cells)/(an inhibitory protein bound to NFκB). After the degradation of the NF-κB/IKB dimer, NFκB complex is translocated to the nucleus where the gene transcription of many pro-inflammatory cytokines, tumor necrosis factor-alpha (TNFα), and interleukin occur [145]. As previously mentioned, TLR4 stimulation by Gram-negative bacteria causes enterocyte death and mucosal injury, both of which have been related to the etiology of NEC in several studies. 

It has been shown that *Bifidobacterium* probiotics and their metabolites can alter the transcriptional activity of enterocytes and modulate the intestinal innate immune response. For instance, probiotic-conditioned media (PCM) with a single probiotic strain or combined probiotic strains including *B. infantis* and *L. acidophilus* could lead to a significant decrease in the expression of IL-1β, IL-8, IL-6, TLR2 mRNA, and TLR4 mRNA and high expression of inflammatory inhibitors (Tollip and SIGIRR). Exposure of PCM with primary enterocyte cultures of NEC tissue has also led to down-regulation of IL-6, IL-8, and TLR2 and up-regulation of Tollip and SIGIRR [146]. 

Similar to this, transcription profiling of immature human fetal intestinal epithelial cells exposed to *B. infantis* and *L. acidophilus* revealed modification of several genes involved in immune responses and cell survival pathways. Probiotic conditioned media (PCM)-exposed cells displayed decreased NF-B pathway gene expression as well as IL-6 and IL-8 levels. As a result of PCM exposure, genes involved in remodeling the extracellular matrix were also downregulated [147]. Given the strong influence of probiotic strains on the regulation of NF-κB pathways, it can be a potential therapeutic strategy to manipulate receptors and cytokines which leads to the activation of this pathway in the functionally immature intestinal tract in preterm infants. TLR2 detected on the surface of several immune cells has also the same function as TLR4. Since immature enterocytes in preterm infants have been associated with high expression of TLR2, probiotic administrations have shown a significant impact on the regulation of TLR2-ligand interaction. The heterodimeric complex of TLR2 with TLR1 and TLR6 can recognize Gram-positive bacteria compounds such as lipoteichoic acids, peptidoglycan, and lipopeptides. The interaction of TLRs and microbial signals leads to the activation of a cascade of immune responses [148]. 

Modulation of TLR2 and TLR4 expression and development of the immune system through probiotic strain activities have been investigated in several animal-based studies. For instance, an investigation of *Bifidobacterium* administration in intestinal epithelial cells in rat models showed that TLR2 expression was significantly lower in intestinal epithelial cells treated with different strains of *Bifidobacterium* (*B. longum*, *B. infantis*, and *B. youth*). While cells infected by *E. coli* endotoxin showed higher expression of TLR2 and TLR4. Also, intestinal barrier function measured by transepithelial/transendothelial electrical resistance (TEER) was significantly higher in *Bifidobacterium*-treated cells compared to cells infected by *E. coli* endotoxin [149].

Human cytokine synthesis inhibitory factor (CSIF) or interleukin 10 (IL-10) mainly produced by monocytes and other immune cells such as Th2, Treg, mast cells, and B cells, is another anti-inflammatory cytokine that can be regulated by probiotics. IL-10 can suppress the production of several pro-inflammatory cytokines including TNFα, IFN-γ, GM-CSF, IL-2, and IL-3 [150]. Animal-based studies have also confirmed the regulatory effect of *Bifidobacterium* strains on IL-10 and subsequently the prevention of inflammatory bowel diseases. For instance, *L. casei* and *B. breve*-treated mouse models could selectively enhance the amount of IL-10-producing CD4+ T cells in the large intestine by twofold without altering intestinal microbiota [151]. *B. adolescentis* supplementation in preterm rat models could also decrease the development of NEC through the modulation of inhibitory adaptor proteins such as TOLLIP, and inhibitory receptor toll interleukin-1R 8 (SIGIRR) and expression of TLR4 [152]. 

Inhibition of NLRP3 inflammasome (NOD-, LRR- and pyrin domain-containing protein 3) has also shown promising results in the prevention of gastrointestinal disorders. NLRP3 inflammasome is a cytosolic multiprotein oligomer in the innate immune system belonging to the nucleotide-binding oligomerization domain-like receptors (NOD-like receptors: NLRs). 

NLRP3 acts as a pattern recognition receptor (PRR) and can detect microbial signals and lead to the production of proinflammatory cytokines (IL-1β) and caspase 1 [153]. Overactivation of NLRP3 has been associated with the development of different inflammatory diseases which can be regulated by the inhibitory effects of probiotic strains [154]. Investigation of NLRP3 inflammasome in NEC mouse models treated with NLRP3 inhibitor MCC950 showed that the NEC mouse model showed higher expression of NLRP3 in the intestine and brain and mature IL-1β compared to mice receiving NLRP3 inhibitor (MCC950). As a result, inflammatory cytokines, NEC survival rate, and histological damage in the brain and gut were all dramatically decreased by MCC950 treatment, demonstrating the significance of blocking the NLRP3 pathway in the prevention of inflammatory bowel disorders [150].

### 6.3. Regulation of Intestinal Epithelium Function

The gastrointestinal barrier provides a vast surface for interacting with microbial signals and environmental stimuli. This contact has a substantial impact on the host’s physiology and may trigger a regulated and normal immunological response or infection development, depending on the initial stimulus. The outermost layer of the intestinal epithelium is made up of enterocytes, Paneth cells, goblet cells, intraepithelial lymphocytes, and enteroendocrine cells. While controlling permeability and microbial translocation, epithelial tight junctions (TJs) between intestinal cells maintain the integrity of the intestinal barrier [155]. Increased intestinal permeability, TJ disruption, and subsequent uncontrolled translocation of microbial pathogens (leaky gut) may occur in preterm newborns with an underdeveloped gut barrier and lead to gastrointestinal diseases. 

Human and animal trials have shown the prophylactic effects of *Bifidobacterium* strains on the intestinal barrier [156]. Investigation of *Bifidobacterium’s* role on the TJ and intestinal barrier in animal models and human intestinal cell models (Caco-2) has shown that *Bifidobacterium* administration can down-regulate the expression of proinflammatory cytokines and improve transepithelial electrical resistance and permeability of Caco-2. *Bifidobacterium* in 10^8^ CFU could also increase the expression of ZO-1, occludin, and claudins (TJ proteins) (*p* < 0.01) compared to Caco-2 monolayers treated with LPS. Moreover, compared to the LPS-induced enterocyte barrier injury of Caco-2 monolayers (*E. coli 055*), and LPS-fed mice models, *Bifidobacterium* significantly suppressed the expression of TNF-α and IL-6 and decreased the NEC rate from 88 to 47% (*p* < 0.05) in controls [157]. Another study showed a different approach in regulation of intestinal barrier by *Bifidobacterium* strains. This study has demonstrated that *B. bifidum* (10^8^ CFU) might improve the intestinal epithelial tight junction barrier in Caco-2 monolayers by targeting the TLR2 pathway in an NF-B-Independent manner (attachment with enterocyte TLR-2 receptors and stimulation of p38 kinase pathway) [158].

### 6.4. Competitive Exclusion and Adhesion Properties

Elimination of pathogens with identical needs for resources by probiotic strains known as competitive exclusion is a common strategy applied by probiotic microorganisms in the gastrointestinal tract [159]. Adhesion of probiotics to the intestinal epithelium can prevent the attachment and colonization of bacterial pathogens, especially enteropathogens, and resultant infections. Probiotics adhesion can also enhance host-probiotic interaction which leads to longer transient colonization time and provide sufficient time to express their immunomodulatory effects while attached to the epithelial receptors [160]. 

Serine protease inhibitor (serpin) produced by *B. longum* subsp. *Longum NCC2705, B. longum* subsp. *Infantis, B. dentium*, and *B. breve* and pentapeptide (CHWPR) in *B. animalis* are common extracellular proteins that facilitate host-probiotic interaction. Neutrophil and pancreatic elastases which are produced during inflammation by immune cells can be prevented by *Bifidobacterium* serin and suppress inflammatory responses and immune cell recruitment [161]. CHWPR can also pass through the cytoplasmic membrane and reach the nucleus and upregulate c-myc and il-6 genes, which are involved in many cellular metabolisms including gastrointestinal tract physiology [162]. 

Several in vitro and in vivo studies have investigated different extracellular proteins in probiotics using intestinal cell lines to evaluate the antagonistic interactions between pathogens and probiotics. In a study assessing the adhesion ability of 12 commonly used probiotic strains and antagonistic interactions with enteropathogens (*Enterobacter*, *Clostridium*, *Staphylococcus*, and *Bacteroides*), all tested probiotic strains could prevent bacterial pathogen colonization in the intestinal epithelium models [163]. Tight adhesion (Tad) pili (Type IVb pili) in *B. breve UCC2003* has also been found to be a critical element for gut colonization (202) and has a proliferation impact on intestinal epithelial cells in mice models [164]. A comparative study on the physiological characteristics and acid-resistant phenotype of *B. longum* and *B. catenulatum* has shown that acid-resistant *Bifidobacterium* strains showed a greater adhesion to the human intestinal mucus and a higher displacement ability (competitive exclusion) on *E. coli*, *Salmonella enterica serovar Typhimurium*, *Listeria monocytogenes*, *Enterobacter sakazakii*, and *Clostridium difficile* from adhering to human intestinal mucus compared to the acid-sensitive strains. These results highlight the significance of carefully evaluating the safety, effectiveness, and phenotypic traits of probiotic strains before clinical trial research [165]. 

The human plasminogen-binding activity of different species of *Bifidobacterium* (*B. bifidum*, *B. longum*, and *Bifidobacterium lactis*) has also shown that *Bifidobacterium* has a unique adhesion ability through degradation of the extracellular matrix which allows *Bifidobacterium*-host interaction [166]. 

In another study, the phenotypic characteristics of *B. breve* and *B. longum* isolated from preterm and full-term infants were examined. This study revealed a significant variation across different isolates in terms of Caco-2 cells adhesion, surface hydrophobicity, and autoaggregation properties which may show strain-specific phenotypic traits that should be considered when choosing the probiotic candidate for modifying the gut microbiota in preterm newborns [98]. It might also explain why, despite probiotic treatment, some investigations have not shown successful competitive exclusion or fecal detection of *Bifidobacterium*. These findings may point to the need for a case-by-case comparison of probiotic strains and infectious agents in order to identify the optimal probiotic candidate with the potential to adhere to and colonize the gastrointestinal tract while also improving disease outcomes.

### 6.5. Synthesis of Antimicrobial Compounds

Another successful tactic against Gram-positive and Gram-negative bacteria is the production of antibacterial compounds by probiotic strains. There have been several low molecular weight compounds (LMWs) found in *Bifidobacterium* strains that show inhibitory properties against pathogens. For instance, short-chain fatty acids (such as acetate, butyrate, and propionate) are the end-products of the metabolism of human undigestible carbohydrates produced by gut microbiota and probiotic strains and have been used as health indicators in the diagnosis of gastrointestinal diseases. In multiple human and animal investigations, the administration of *Bifidobacterium* was linked to greater levels of short-chain fatty acids and a reduction in intestinal damage [85,90,167]. Numerous studies have also linked LMW lipophilic compounds to the inhibitory actions of *Bifidobacterium* [168,169]. In Caco-2 cells and mouse models, for example, the antibacterial activity of 14 *Bifidobacterium* strains isolated from newborn fecal samples against *S Typhimurium SL1344* revealed antagonistic action of *Bifidobacterium* strains either through cell entry prevention or intracellular inhibition [168].

## 7. Safety of *Bifidobacterium* Probiotic

Despite the common use of probiotics in preterm infants and the low rate of adverse effects, controversies remain around the safety, short-term and long-term effects of probiotic administration. The safe use of probiotics in preterm infants has been documented in numerous studies, but there is no guarantee of their absolute safety, which calls for ongoing observation and case-by-case evaluation. 

For instance, in a preterm infant with surgery for omphalocele four hours after birth and treated with *Bifidobacterium breve BBG-01* probiotic on day 2, the blood culture was positive for *Bifidobacterium breve BBG-01* (resistant to meropenem, and susceptible to Ampicillin/Sulbactam and penicillin in vitro), which was genetically identical to orally administered probiotic strain [170]. This may raise the importance of case-by-case evaluation and potential risk factors of probiotic strains in preterm infants with particular medical conditions as other preterm infants had been treated with a similar probiotic strain without displaying any systematic consequence in this study. Another case study reported LOS diagnosis in a preterm infant with laparotomy and probiotic treatment (*Lactobacillus rhamnosus*). It is important to note that in these two case studies, both infants were diagnosed with underlying intestinal diseases prior to probiotic treatment [171].

There are still a lot of unanswered questions surrounding the target population, the choice of efficient probiotic strains, the length of therapy, and the dosage. For instance, a systematic review and meta-analysis of 51 randomized controlled trials with 11231 preterm infants revealed that not all research used the same probiotic strains and same dosage in preterm newborns, making it challenging to evaluate the safety, effectiveness, and optimal dosage of different probiotic strains. In this systematic and meta-analysis review, three combined probiotic therapy out of 25 studies showed a mortality reduction rate, seven therapies decreased NEC, two reduced LOS, and three treatments reduced enteral feeding time [172]. However, this study was unable to draw any definitive conclusions regarding the most effective probiotic strains for various clinical outcomes which might be due to a limited number of studies and lack of a standardized method of probiotic treatment. Another meta-analysis review with 24 studies showed a significant association between probiotic administration and NEC reduction rate and mortality, with no remarkable impact on LOS and without any reported systematic infection after using a single probiotic strain (*lactobacillus*) or in combination with *Bifidobacterium* strains [173]. These findings show that a lack of adherence to a standard protocol in probiotic therapy could result in the inappropriate or even unsafe administration of probiotics to premature infants.

From the manufacturing point of view, probiotic strains undergo five main phases: strain selection, culture, fermentation, centrifuge, and blending. As probiotic manufacturing has a long history in the food industry, probiotic strains are mainly certified for dietary use as a food supplement, not for medical purposes which is of the utmost importance, particularly in vulnerable individuals with medical conditions. Also, the Qualified Presumption of Safety provided by the European Food Standards Agency (EFSA) and the Good Manufacturing Practice (GMP) provided by the Food and Drug Administration (FDA) in the USA do not require the medical efficiency and quality of probiotic strains [174]. Additionally, some beneficial properties of probiotics are just strain-specific and shouldn’t be generalized to a formulation. Potential phenotypic and genotypic variations in probiotic strains under in vivo and in vivo conditions should also be noted when determining the beneficial effects of probiotic strains [174]. For example, a comparison of *Bifidobacterium* strains in 16 probiotic products showed that in over 90% of the cases, *Bifidobacterium* strains in the product did not show the same descriptions and properties claimed on the package label [175,176]. These findings highlight the urgent need for a regulated and consistent protocol from probiotic strain production to delivery, especially for medical uses. 

Though there are not many case studies reporting the adverse effect of *Bifidobacterium* probiotics, the potential risks should not be discounted because some adverse effects may remain unreported due to difficulty in isolation of probiotic strains which are usually anaerobes and hard to grow. Given the difficulties in isolating probiotic strains from clinical specimens, research and diagnostic laboratories should be equipped with proper methods and tools to accurately evaluate probiotic strains. Benefits and risk considerations should be assessed in critically ill populations, even though a consistent protocol can enhance benefits and decrease adverse effects. Also, studies utilizing different probiotic strains should take further measures to identify, assess, and report any relevant risk factors. 

## 8. Conclusions

*Bifidobacterium* is one of the initial and dominant colonizers of the gastrointestinal tract with protective and immunomodulatory roles. Many preclinical and clinical studies have shown the effectiveness of *Bifidobacterium* probiotics as a therapeutic approach in the prevention and treatment of preterm infant complications including inflammatory intestinal diseases, neurodevelopmental diseases, and allergies. However, many studies discussed here had limitations, including a possible bias in the study design, small sample size, cross-contamination, low follow-up rate, single-center comparison, and lack of a standardized method in terms of probiotic dose and treatment duration. Therefore, well-designed studies with larger sample sizes are required to fully evaluate the reciprocal interaction between the host and *Bifidobacterium* probiotic. Also, further investigations in human and animal trials are needed to fully evaluate the effectiveness of *Bifidobacterium* (single or combined probiotics) as a microbiome-targeted intervention for the re-programming of the gut microbiota and treatment of gut-microbiota-associated diseases in preterm infants and other vulnerable populations.

## Figures and Tables

**Figure 1 nutrients-15-00709-f001:**
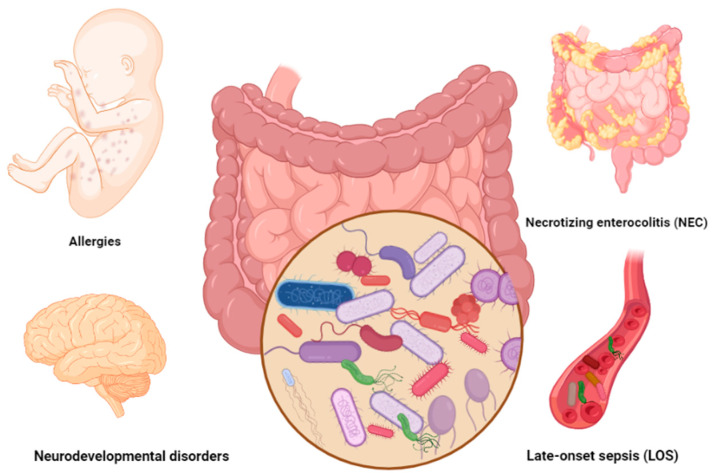
Common gut-microbiota-associated complications in preterm infants. Figures in this manuscript were created specifically for this manuscript in BioRender.com, accessed on 15 August 2022.

**Figure 2 nutrients-15-00709-f002:**
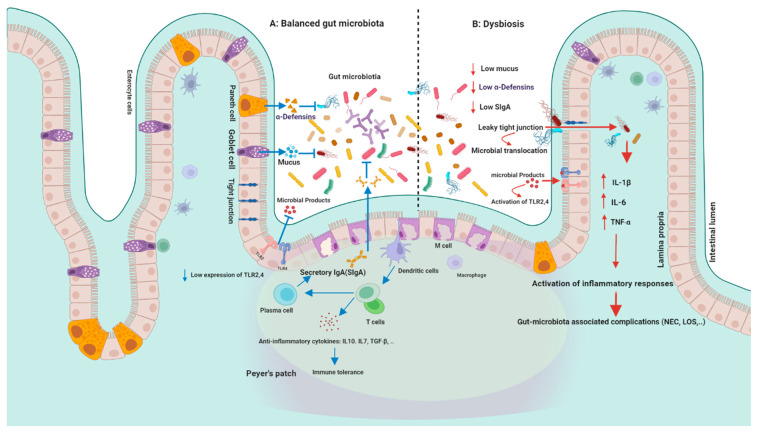
Gut microbiota and immunity: (**A**) in healthy conditions: intestinal epithelial cells provide a protective layer between intestinal mucosa and luminal microorganisms. Goblet and Paneth cells secrete mucus layer and antimicrobial peptides. Secretory immunoglobulin A (sIgA) secreted by B cells have protective roles against luminal microbiota. While controlling permeability and microbial translocation, epithelial tight junctions (TJs) between intestinal cells maintain the integrity of the intestinal barrier. Gut microbiota-immunity crosstalk can activate different immunological pathways either in a regulated or exaggerated way and lead to the development of several diseases. Host-microbe interaction is activated through different recognition receptors, which are highly expressed in intestinal epithelial cells (IECs) such as TLRs. PRRs activation can lead to the production of different antimicrobial peptides such as α-defensins. Host-microbe interaction can influence T cell differentiation into Th1, Th2, Th17, and Treg cells, which are regulated by pro-inflammatory and anti-inflammatory cytokines such as TGF β and IL-10. (**B**) Dysbiosis; recognition of microbial compounds (such as Gram-negative lipopolysaccharide) by TLRs leads to the activation of MYD88 and the production of several inflammatory cytokines. TLR4 stimulation by Gram-negative bacteria causes enterocyte death and mucosal injury. TLR2 (TLR1 and TLR6) can also recognize Gram-positive bacteria. The interaction of TLRs and microbial signals leads to the activation of a cascade of immune responses. Increased intestinal permeability, TJ disruption, and subsequent uncontrolled translocation of microbial pathogens (leaky gut) can lead to several gastrointestinal diseases. Figures in this manuscript were created specifically for this manuscript in BioRender.com, accessed on 15 August 2022.

**Figure 3 nutrients-15-00709-f003:**
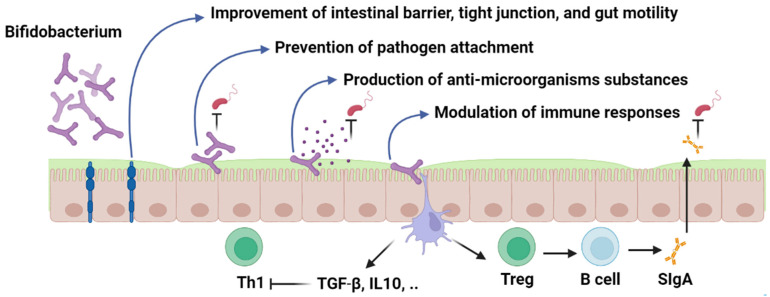
*Bifidobacterium*; mechanism of action. Figures in this manuscript were created specifically for this manuscript in BioRender.com, accessed on 15 August 2022.

**Table 1 nutrients-15-00709-t001:** Clinical trials on the effects of *Bifidobacterium* strains in preterm infants.

Reference	Year	Location	Study Design	Aim	Sample Size	Recruiting Center	Inclusion Criteria	Exclusion Criteria	Gestational Age	Weight	Probiotic/Case Group	Probiotic Dose	Placebo/Control Group	Grouping Assignment	Treatment Duration	Disease	Probiotic Safety	Limitation (As Described in the Study)	Conclusion
[83]	1997	Osaka_ Japan	randomized controlled trial	Impact of *Bifidobacterium breve* YIT4010 (BBG) supplementation on fecal counts and possible adverse effects	116 (66 probiotic, 50 placebo)	1 neonatal intensive care unit	birthweight of under 1500 g	major anomalies, severe asphyxia, severe intrauterine growth retardation	≅28.20	<1000 g	*B breve YIT4010*	0.5 × 10^9^	distilled water	randomly allocated	Daily single dose from initiation of milk feeds to 28 days		no side effects	Possible cross-contamination of placebo and probiotic groups	Effective colonization of *B. breve*, Probiotic association with less abnormal abdominal signs and better weight gain
[84]	2004	Tokyo_ Japan	controlled trial	Evaluation of *Bifidobacterium breve* impact on intestinal flora and fecal Bifidobacterium abundance	30 (20: probiotic (10: received probiotic several hours after birth (group A), 10: received probiotic 24 h after birth (group)), 10: placebo)	1 neonatal intensive care unit	admitted to the Neonatal Intensive care unit of Juntendo University Hospital between 2000 and August 2002	deformities, chromosomal abnormalities, or intrauterine intrauterine infection	≅32.8 weeks	from 780 to 2250	*Bifidobacterium breve*	1.6 × 10^8^ cells	fed normally without supplement	subjects were randomly divided into three groups	twice a day at the time of normal feeding continued until discharge	Respiratory Distress Syndrome, NEC, sepsis	No side effect	ND	Infants with early *Bifidobacterium* administration had significantly earlier detectable *Bifidobacterium* count
[85]	2007	Tokyo_ Japan	ND	*Bifidobacterium breve M-16V* supplementation impact on fecal lactic acid and short-chain fatty acids (acetate, propionate, and butyrate acids)	66	1 neonatal intensive care unit, 1 hospital	ND	malformations, chromosomal abnormalities, or intrauterine infections	<36 weeks	<2500	*B breve M-16V* (Morinaga Milk Industry, Kanagawa, Japan)	1.6 × 10^8^ cells	no supplement	randomly divided into probiotic an placebo groups	first day of life irrespective of the use of enteral feeding twice daily until discharge	ND	ND	ND	A lower concentration of fecal acetic acid and butyric acid was detected after probiotic administration which may have protective roles against digestive diseases
[86]	2006	Potsdam_ Germany	double-blind, placebo-controlled, randomized study	Effect of *Bifidobacterium lactis* Bb12 on gut microbiota	69 (37 probiotic, 32 placebo)	1 hospital	ND	chromosomal aberration, human immunodeficiency virus infection in the mother, hydrops fetalis, and inborn malformation of the gastrointestinal tract	From 30 to 35 weeks	990 to 2750 g	*Bifidobacterium lactis Bb12*	1.6 × 10^9^ cells on day 1 to 3 and 4.8 × 10^9^ cells from day 4 onward	formula-based placebo	Randoma software version 4.3	first day after birth and continued for 21 days	ND	ND	ND	A higher fecal abundance of *Bifidobacterium* in the probiotic group, a lower abundance of *Enterobacteriaceae* and *Clostridium* spp. in probiotic group
[87]	2007	Athens_ Greece	prospective randomized case-control	Investigation of the role of probiotic administration on intestinal permeability, growth, sepsis, and NEC rate	75 (41 probiotic, 34 placebo)	1 hospital	gestational age between 27 and 37 weeks, stable state, formula-fed	major deformities, such as congenital heart defects or bowel atresia	<36 weeks	<1500	formula supplemented with *Bifidobacterium lactis* (Nestlé, Vevey)	2 × 10^7^ cfu/g of dry milk	same formula without probiotic	randomly assigned (balanced block randomization)		NEC, sepsis	ND	small sample size	Probiotic administration well-tolerated and decreased intestinal permeability and led to increased head growth and *Bifidobacterium* count
[88]	2006	Tokyo_ Japan	randomized controlled study	The role of *Bifidobacterium Breve* administration on transforming growth factor A1 signaling (TGF-A1)	19 (11: probiotic, 8: placebo)	1 neonatal intensive care unit	ND	chromosomal or congenital anomalies or history of intrauterine infection or surgery, Infants who had received or whose mothers had received corticosteroid treatment	<36 weeks	<2500	*B. breve M-16V* g: live but not viable bacteria (Morinaga Milk Industry, Kanagawa, Japan)	1 × 3 × 10^9^ CFU	5% glucose solution (without any B. breve)	allocated to 1 of 2 groups	starting several hours after birth twice a day	NEC, Respiratory distress syndrome, Infection, Retinopathy of prematurity, Chronic lung disease	No adverse effect	limited analysis of peripheral samples rather than more elements of the mucosal immune system	*B. breve* administration could up-regulate TGF-A1 signaling which has anti-inflammatory and allergic responses
[89]	2007	France	prospective study	Colonization of *Bifidobacterium* in preterm infants	52	2 hospitals	infants with gestational age ranging from 30 to 35 weeks hospitalized in the neonatal intensive care unit	deformities, chromosomal abnormalities, or inappropriate weight for gestational age	From 30 to 35 weeks	990 to 2750 g	all infants received standard formula (with 2 probiotic strains (ie, *B. breve C50* and *Streptococcus* thermophilus)) with mother milk	ND	all infants received standard formula (with 2 probiotic strains (ie, *B. breve C50* and *Streptococcus* thermophilus)) with mother milk	ND	ND	ND	ND	ND	Gestational age had a significant impact on *Bifidobacterium* colonization and gut maturation
[90]	2008	Potsdam_ German	double-blind placebo-controlled randomized prospective clinical trial	Effects of *Bifidobacterium lactis* Bb12 Supplementation on indicators of health status (fecal pH, acetate, lactate, calprotectin, IgA, and body weight)	69 (37 Probiotic, 32 Placebo)	1 hospital	German ethnic background (except one Russian background)	ND	<37 weeks	<1500	*Bifidobacterium lactis*	1.6 × 10^9^ cells per g of powder	human milk fortifier	randomly assigned	daily for 21 days	ND	ND	ND	Early probiotic supplementation resulted in higher body weight, higher concentrations of fecal acetate, lactate, and IgA, and lower fecal pH. Fecal calprotectin was lower in the probiotic group
[91]	2008	Taipei_ Taiwan	prospective, blinded, randomized, multicenter controlled trial	Investigation of the efficacy of *Bifidobacterium bifidum* and *Lactobacillus acidophilus* probiotics in prevention of NEC	434 (217 probiotic, 217 placebo)	7 neonatal intensive care units	Very low birth weight infants who survived to start enteral feeding	severe asphyxia (stage III), fetal chromosomal anomalies, cyanotic congenital heart disease, congenital intestinal atresia, gastroschisis, or omphalocele, infants with exclusive formula feeding, and those who were fasted for more than 3 weeks	<34 weeks	<1500	*Bifidobacterium bifidum* and *Lactobacillus acidophilus* (Infloran)	10^9^ colony-forming units	breast milk or mixed feeding	assigned randomly to 2 groups using sequential numbers generated at the computer	twice daily, with breast milk or mixed feeding (breast milk and formula) for 6 weeks	NEC (stage ≥ 2)	No adverse effect	ND	NEC rate and mortality were lower in probiotic group
[92]	2010	Ulm_ Germany	randomized Controlled Trial	Investigation of the role of *Bifidobacterium lactis* in prevention of nosocomial infections	103 (93 probiotics, 90 placebo)	1 Children’s Hospital	infants < 30 weeks of gestation admitted to the Division of Neonatology (Children’s Hospital, University of Ulm, Germany)	early death, congenital malformations or missing parental consent	<30 weeks	<1500	*Bifidobacterium lactis*	6 × 2.0 × 10^9^ CFU/kg/day, 12 billion CFU/kg/day	Human milk fortifier powder	randomly assigned (sealed envelopes, computer-generated, blocked randomization lists, block size of four)	ND	NEC, nosocomial infection	No adverse effect (blood culture)	ND	Probiotic administration did not have a significant impact on nosocomial infection prevention and NEC incidence, *B. lactis* administration had no adverse effect
[93]	2011	Recife_ Brazil	prospective, double-blind, randomized, controlled trial	*Bifidobacterium breve* and *Lactobacillus casei* supplementation impact on NEC	231 (119 probiotic, 112 placebo)	1 neonatal intensive care Unit	Infants born locally and admitted to the Neonatal Intensive Care Unit (NICU) with a birth weight from 750 to 1499 g	major congenital malformations, life-threatening chromosomal alterations, and/or congenital infections	≅29.35 weeks	750 to 1499 g	*L. casei* and *B. breve*	3.5 × 10^7^ to 3.5 × 10^9^ CFU (Yakult LB, Brazil)	human milk	randomly assigned using a randomization program (Epi-Info 6.04)	first month of life	NEC (stage ≥2)	no side effects	ND	Probioitc reduced NEC, improved intestinal motility
[94]	2012	Tokyo_ Japan	prospective randomized control trial	Early *Bifidobacterium bifidum* OLB6378 supplementation impact on growth and morbidity	36 (early probiotic supplementation (within 48 h of birth), late probiotic supplementation (more than 48 h after birth))	1 maternal and perinatal Center	ND	major congenital malformations, systemic infection, and the lack of parental consent, anticipated feeding problem	<30 weeks	<1500	*B. bifidum*/500 mg (Meiji, Tokyo, Japan)	2.5 × 10^9^ viable cells	ND	random-number generation and a 1:1 allocation	Daily until the bodyweight reached 2 kg	ND	reported safe (monitoring for sepsis with positive blood culture, the length of hospital stay, and the level of B.bifidumin in the fecal samples)	small sample size	Early administration of probiotics had a significant impact on growth (daily body weight gain) and mortality, No significant differences were found in the fecal *Bifidobacterium* level between the groups (However, it was higher when the supplementation started between 24 and 48 h after birth),
[95]	2013	Tokyo_ Japan	ND	Comparing single and combined probiotic strains on *bifidobacterial* abundance	44 (probiotic (15 one species group, 13 three species group), 16 placebo)	1 neonatal intensive care unit		infectious diseases, infants treated with antibiotics	<34 weeks	<2000	*Bifidobacterium breve M-16V* (one-species group), *B. breve M-16V, Bifidobacterium longum* subsp. *infantis M-63* and *B. longum* subsp. *longum BB536* (mixture of three specie)	5 × 10^8^ (one-species group), 5 × 10^8^ (of each strain; three-species group)	ND	ND	Daily from the beginning of enteral nutrition for 6 weeks		ND	ND	Significant increase of *Bifidobacteria* count in the probiotic group, Three species probiotics resulted in earlier formation of bifidobacterial colonization, Lower abundance of Clostridium in the probiotic group, Lower *Enterobacteriaceae* abundance in the three species, the proportion of bifidobacteria in the three-species group was significantly higher than that in the one-species group, *B. breve* M-16V and *Bifidobacterium infantis* M-63 were detected in majority of infants
[96]	2013	Melbourne_ Australia	prospective multicenter, double-blinded, placebo-controlled, randomized trial	Evaluation of the impact of Probiotics (*B. infantis, S. thermophilus, and B. lactis*) on Los	1099 (548 probiotic, 551 placebo)	10 perinatal hospitals	infants with <32 weeks gestational age and weighing <1500 g within 72 h of birth	major congenital or chromosomal anomalies, if death was considered likely within 72 h of birth if mothers were taking nondietary probiotic supplements	<32 weeks	<1500	*Bifidobacterium infantis, Streptococcus thermophilus*, and *Bifidobacterium lactis* (ABC Dophilus Probiotic Powder for Infants; Solgar, Leonia, New Jersey)	300 × 10^6^ (*Bifidobacterium longum* subsp. *Infantis BB–02*), 350 × 10^6^ (TH–4, *Streptococcus thermophilus*), 350 × 10^6^ (*Bifidobacterium animalis* subsp. *Lactis BB-12*)	maltodextrin	1:1 allocation using STATA	Daily until discharge from hospital or term corrected age	NEC, Los	reported safe	ND	A significant decrease in NEC rate (stage 2 or more) (but not sepsis, and all-cause mortality) was observed after probiotics administration, No probiotic adverse effect was reported
[97]	2014	Perth_ Australia	randomized double-blinded placebo-controlled trial	Role of *Bifidobacterium breve* (B. breve) M-16V supplementation on fecal counts and possible adverse effects	159 (79 Probiotic, 80 Placebo)	1 tertiary neonatal intensive care unit	Infants with <32 weeks and 6 days, under 1500 g, ready to commence or on enteral feeds for <12 h	Major congenital malformation, chromosomal aberration, lack of informed parental consent, enteral feeds for ≥12 h, contraindications for enteral feeds, life-threatening illness	<33 weeks	<1500 g	B. breve M-16V	3 × 10^9^ cfu/day	dextrin	randomly allocated	Daily single dose until the corrected age of 37 weeks	NEC (≥Stage 2)	no side effects: blood culture for B. breve M-16V, monitoring adverse effects such as abdominal distension, vomiting, and diarrhea	Immediate supplementation of probiotic without considering B. breve counts in meconium	Routine use *of B. breve M-16V* is safe, No adverse effect was reported, Probiotics significantly increased *B. breve* fecal count
[98]	2014	France	ND	Comparison of cell surface properties (autoaggregation, hydrophobicity, and Caco-2 cells adhesion) of *B. longum* and *B.breve* isolates in preterm and full-term infants	47 (20 preterm, 27 full term)	ND	ND	ND	<36 weeks	ND	ND	ND	ND	ND	ND	ND	ND	ND	Cell surface properties were different between *Bifidobacterium* strains isolated from preterm and full-term infants
[99]	2016	London_ UK	multicentre, randomized controlled phase 3 study	Evaluation of *Bifidobacterium breve BBG-001* supplementation impact on NEC, sepsis	1310 (650 probiotic, 660 placebo)	24 hospitals	ND	Infants with a potentially lethal malformation or any malformation of the gastrointestinal tract apparent by 48 h and those with no chance of survival	From 23 and 30 weeks	<1000 g	B breve BBG-001 (Yakult Honsha Co Ltd.)	enterally in a daily dose of 8·2 to 9·2 log10 CFU	corn starch	based on date of birth	Daily single dose until the infant reaches a corrected post-menstrual age	NEC (Bell’s stage 2 or 3)	no short-term safety	Possible cross-contamination of placebo and probiotic groups	Probiotic did not have any significant impact on NEC and sepsis,
[100]	2015	Viçosa, Brazil	pilot study	Evaluation of *Bifidobacterial* composition in full-term and preterm infants	49 (24 full term, 25 preterm)	1 hospital	availability of fecal samples and signed informed consent from the mother. Being residents of Viçosa and neighboring areas	ND	<39 weeks	<3500	one-month-old preterm infants	ND	one-month-old full-term infants	ND	ND	ND	ND	ND	*Bifidobacterium longum* colonized in all full-term and pre-term newborns. Variation in fecal counts of *Bifidobacterium* genus and *Bifidobacterium longum* between full-term and pre-term infants. Variation of *Bifidobacterium lactis* abundance between preterm cesarean and pre-term vaginally born infants.
[101]	2015	Turkey	multicenter, prospective, randomized, double-blind, randomized controlled trial (RCT)	Investigation of the prevention role of probiotics and prebiotics (alone or combined (synbiotic)) on necrotizing enterocolitis	400 (100: probiotic, 100: prebiotic, 100: synbiotic, 100: placebo)	5 neonatal intensive care units	gestational age of <32 weeks and a birth weight of <1500 g, born at or transferred to the NICUwithin the first week of life and fed enterally before inclusion	Infants with any disease other than those linked to prematurity or congenital anomalies of the intestinal tract, not fed enterally or who died before the seventh day after birth, whose mothers had taken nondietary probiotic supplements, and whose parents refused to participate	<32 weeks	<1500	probiotic (*Bifidobacterium lactis*), prebiotic (inulin), synbiotic (*Bifidobacterium lactis*)	probiotic (5 × 10^9^ colony-forming units), prebiotic (900 mg), synbiotic (5 × 10^9^ colony-forming units probiotic, 30 mg plus inulin, 900 mg)	breastmilk or formula without the addition of probiotic or prebiotic and received maltodextrin	randomly assigned (balanced blocks using sealed envelope)	maximum of 8 weeks before discharge or death (variables between groups)	Bell stage II-III, bronchopulmonary dysplasia, intraventricular hemorrhage, cystic periventricular leukomalacia, and retinopathy of prematurity		findings cannot be generalized to all probiotics with different doses, limited inclusion to infants who survived beyond the 7 days of life	Probiotic (*Bifidobacterium lactis*) and synbiotic (*Bifidobacterium lactis* plus inulin), but not prebiotic (inulin) alone could decrease the NEC rate in the probiotic group
[102]	2016	Perth_ Australia	retrospective cohort study	*Bifidobacterium breve M-16V* supplementation impact on NEC	1755 (920 probiotic, 835 placebo)	ND	preterm neonates born <34 weeks	major congenital malformations, chromosomal aberrations, and contraindications for enteral feeding, and those with no informed consent	<34 weeks	≅1340	*Bifidobacterium breve M16V*	3 × 10^9^ (3 billion) cfu/day	ND	ND	Daily single dose continued until the corrected age of 37 weeks	NEC (stage ≥2)	no adverse effects monitoring for sepsis and abdominal distension, vomiting, and diarrhea	It was a retrospective design, which made it difficult to control all confounders	*Bifidobacterium breve M-16V* was associated with reduced NEC and mortality
[103]	2016	Perth_ Australia	analysis of a randomized trial	Impact of *Bifidobacterium breve* M-16V supplementation on fecal *Bifidobacterium*	153 (77 probiotic, 76 placebo)	1 tertiary neonatal intensive care unit	Preterm infants with small for gestational age due to small for gestational age	chromosomal aberrations, congenital malformation	<33 weeks	<825	*B. breve M-16V*	3 × 10^9^ cfu/day	dextrin	randomly allocated	until the corrected age of 37 weeks	ND	reported safe (by monitoring blood culture positive sepsis by B. breve M-16V and adverse effects such as abdominal distension, vomiting, and diarrhea leading to the cessation of the supplementation)	ND	*B. breve M-16V* supplementation did not change the detectable *B. breve* counts between infants with small gestational age (SGA) and non-SGA
[104]	2016	Anhui_ China	ND	Impact of *clostridium butyricum* and *bifidobacterium* (LCBBCP) on the expression of B and T lymphocyte attenuator (BTLA) on CD4 cells	80	1 neonatal intensive care unit	ND	neonatal comorbidities (including asphyxia, infection, congenital malformation, respiratory distress syndrome, pneumorrhagia, congenital immunodeficiency, and other related conditions), maternal infectious diseases during pregnancy and autoimmune disorders	<37 weeks	ND	*clostridium butyricum* and *bifidobacterium* (Changlekang, China)	ND	simple formula milk	equally randomized (random digit table)	twice a day for 7 days	ND	ND	ND	LCBBCP had inhibitory impact on excessive activation of T lymphocytes
[105]	2017	Germany	observational study	Impact of *Bifidobacterium infantis* and *Lactobacillus acidophilus* supplementation on preterm infant growth under antibiotic exposure	8534 (6229 probiotic, 2305 placebo)	54 neonatal intensive care units	birth weight <1500 g, gestational age >22 0/7 and ≤32 6/7 weeks, written informed consent of parents or legal representatives, and discharge to the home environment	lethal malformations, e.g., trisomy 13 and trisomy 18	≤33 weeks	<1500	(Infloran)	10^9^	ND	ND	Daily for 28 days	ND	ND	It was an observational study, not a double-blinded, randomized controlled study, which made it difficult to control confounders and interpret findings properly. For a follow-up cohort, the sample size is not sufficient. Variable duration of hospitalization observed in participants. Bias in designing the study since probiotics were more often given to infants <28 weeks of age	Probiotic supplementation had beneficial impact on weight gain and growth rate in infants under antibiotic exposure
[106]	2017	Melbourne_ Australia	double-blinded, placebo-controlled, randomized trial	Determination of probiotic combination (*B. infantis*, *S. thermophilus,* and *B. lactis*) on neurodevelopmental outcomes in very preterm infants. Follow up	Follow-up 735 (373 probiotics, 362 placeboes)	10 perinatal hospitals	participants in the ProPrems trial	Children for whom disability status could not be determined	<32 weeks	<1500	*Bifidobacterium infantis, Streptococcus thermophilus*, and *Bifidobacterium lactis* (ABC Dophilus Probiotic Powder for Infants; Solgar, Leonia, New Jersey)	300 × 10^6^ (*Bifidobacterium longum* subsp. *Infantis BB–02*), 350 × 10^6^ (TH–4, *Streptococcus thermophilus*), 350 × 10^6^ (*Bifidobacterium animalis* subsp. *Lactis BB-12*)	maltodextrin		Daily until discharge from hospital or term corrected age	NEC, Los	reported safe	declined and lost follow-up participants since the study was not planned as an outcome of the ProPrems trial, a wide age range among participants, which may impact the power of the study to find differences between two groups	No neurodevelopmental and behavioral adverse effect was detected after combined probiotic administration
[107]	2018	Melbourne_ Australia	multi-center, double-blind, placebo-controlled randomized trial	Investigation of the role of *Bifidobacterium infantis BB-02*, *Bifidobacterium lactis BB-12*, and *Streptococcus thermophilus* TH-4 probiotic on gut microbiota composition	66 (38 probiotic, 28 placebo)	1 newborn Intensive Care Unit	infants enrolled at The Royal Women’s Hospital, Melbourne, Australia with at least one swab available	ND	<32 weeks	<1500	Bifidobacterium longum subsp. Infantis, Streptococcus thermophilus, *Bifidobacterium animalis* subsp. *Lactis* (ABC Dophilus Probiotic Powder for Infants; Solgar, Leonia, New Jersey)	300 × 10^6^ (*Bifidobacterium longum* subsp. *Infantis BB–02*), 350 × 10^6^ (TH–4, *Streptococcus thermophilus*), 350 × 10^6^ (Bifidobacterium animalis subsp. Lactis BB-12)	maltodextrin powder	adjusting for age at sampling	once enteral feeds were commenced until discharge or term-corrected age	ND	ND	limited taxonomy classification to the genus level, cross-colonization in the control group, and a small number of ProPrems participants, due to the variable number of samples per infant colonization patterns could not be established for all infants, only 11 specimens collected before supplementation commenced were available (not clear if there was a gut microbial difference between the two allocation groups before supplementation), due to the lower NEC incidence in the selected participants for this study comparison of gut microbiota in NEC and NonNEC infants were not possible	A higher abundance of *Bifidobacterium* in the probiotic group, lower *Enterococcus* abundance in the probiotic group, early BB-02, TH-4, and BB-12 supplementation increased the *Bifidobacterium* abundnace
[108]	2019	Norwich_ UK	single- center retrospective observational study	*Lactobacillus* and *Bifidobacterium* supplementation impact on NEC, sepsis, and mortality	982 (pre-probiotic epoch = 469, routine probiotics = 513)	1 tertiary- level neonatal intensive care unit	<32 weeks’ gestation, plus 32–36 weeks’ gestation VLBW infants. Outborn babies were included if transferred within 72 h of birth	abdominal concerns at referral	<36 weeks	<1500	Initially *Bifidobacterium* and *Lactobacillus* (Infloran capsules), then triple-species Labinic Drops: four drops once daily	Initially 10^9^ colony-forming units (CFU) (*Bifidobacterium* and Lactobacillus) then ~0.5 × 10^9^ CFU dosage each of *L. acidophilus, B. bifidum*, and *B. longum subspecies infantis* daily	donor breast milk (DBM) was available to supplement shortfalls in mother’s own breast milk supply before full feeds. Cow’s milk-based fortifier was added to breast milk between full enteral feeds (≥150 mL/kg/day) and discharge	allocated by date of birth	twice daily on postnatal day 1 until ~34 weeks postmenstrual age	NEC, sepsis	no safety issues	It was a retrospective observational study which made it difficult to control confounders and interpret findings properly	A significant decrease in NEC incidence and sepsis after multispecies probiotic supplementation was observed
[109]	2019	Japan	ND	Impact of probiotic supplementation and timing of initial colostrum on *Bifidobacterium* colonization	98 (group H:37, group L = 30. group N = 31)	1 neonatal intensive care unit	ND	preterm infants without informed consent, congenital malformations	<36 weeks	<2500	group H (received non-live bifidobacteria), and group L (received live bifidobacteria).	Group L: a mixture of 20 mg of live OLB6378 powder (containing 10 mg of lyophilized live OLB6378 concentrate with >2.5 × 10^9^ live cells) and 480 mg of dextrin, Group H: a mixture of 20 mg of lyophilized non-live OLB6378 powder (containing 10 mg of lyophilized non-live OLB6378 concentrate with >2.5 × 10^9^ non-live cells) and 480 mg of dextrin	Group N (no intervention)	ND	within 48 h after birth and continued for at least 1 month after birth	ND	ND	different measurement of bifidobacterial colonization in preterm and term infants, not performing multiple regression analysis	Bifidobacterial colonization in preterm infants at 1 month was low compared to term infants, Higher *Bifidobacterium* colonization was detected after probiotic administration in groups H, and L, Earlier consumption of colostrum had a significant impact on the fecal *Bifidobacterium* count/abundance
[110]	2020	Norwich_ UK	observational longitudinal study	Impact of *Bifidobacterium* and *Lactobacillus* probiotic on fecal metabolites and gut microbiota	234 (101 probiotic, 133 placebo)	4 tertiary-level NICUs	premature infants born at gestational age <34 weeks, and resident in the same NICU for the study duration	necrotizing enterocolitis or severe congenital abnormalities	<28 weeks	<1500 g	Bifidobacterium bifidum, Lactobacillus acidophilus (Infloran, Desma Healthcare, Chiasso, Switzerland):	10^9^ colony-forming units (CFU) of Bifidobacterium bifidum and 10^9^ CFU of Lactobacillus acidophilus	ND	matched by age, sex, and delivery method	Twice daily from the first enteral colostrum/milk feed until 34 weeks post-conceptual age		ND	It was an observational study, not a double-blinded, randomized controlled study, which made it difficult to control confounders and interpret findings properly, not monitor the impact of feeding diet on microbiota in all infants, not measure absolute abundance of bacterial taxa	Association of probiotics with higher abundance of *Bifidobacterium* and higher fecal acetate/lactate concentration, and lower fecal pH
[111]	2020	Perth_ Australia	follow up of a randomized controlled trial	Evaluation of long-term neuropsychological effects of early probiotic supplementation in preterm infants	67 (36 probiotics, 31 placebo)	1 tertiary neonatal intensive care unit	preterm neonates (born <33 weeks) in the original RCT of probiotic for preventing NEC	Major congenital malformation, chromosomal aberration, lack of informed parental consent, enteral feeds for ≥12 h, contraindications for enteral feeds, life-threatening illness	<33 week	<1105	*B. breve M-16V*	3 × 10^9^ cfu/day	dextrin	randomly allocated		Daily single dose until the corrected age of 37 weeks	no side effects: blood culture for B. breve M-16V, monitoring adverse effects such as abdominal distension, vomiting, and diarrhea	low follow-up rate	Probiotics did not have any significant effect on neurodevelopment at the age of 3 to 5 years
[112]	2020	Spain	a prospective and observational study	Impact of donated human milk on gut *Bifidobacterial* profile and metabolism	42 (28 own mother milk, 13 donated milk)	1 hospital	ND	Mixed feeding and use of probiotics, prebiotics, or other treatments. NEC, culture-positive early-onset infection, major malformations, or surgery of the intestinal tract	between 24 and 34 weeks	1334.88 ± 338.64 g (mean ± SD)	human donor milk-fed preterm infants	ND	breastfed preterm infants	ND	At least for the first ten days of life: preterm infants received their own mother’s milk or donated milk. At 30 days of life, half of the babies received formula, with only three babies with their own mother’s milk. the OMM regimen.	ND	ND	mall number of infants and the large interindividual variability, confounding factors, such as antibiotics could influence the findings	A specific bifidobacterial profile was detected based on feeding type. Higher bifidobacterial diversity in the human donor milk group
[113]	2020	Germany	observational study	Impact of *Lactobacillus Acidophilus*, *Bifidobacterium Infantis* probiotic on growth and sepsis	7516	64 neonatal intensive care units	infants with complete documentation for feeding type	lethal malformations or infants treated with comfort (palliative) care	<29 weeks	<1500	Lactobacillus Acidophilus, Bifidobacterium Infantis	1 × 3 × 10^9^ CFU (Colony forming units) L. acidophilus and 1 × 1.5 × 10^9^ B. infantis	exclusively fed with own mother‘s and/or donor‘s milk, fed with HM and formula at any time during the primary stay in hospital, Infants who were exclusively fed with formula	ND	once or twice daily in capsules beginning from day 1 to 3 of life until day 28 of life	sepsis, Bronchopulmonary dysplasia, NEC, focal intestinal perforation, Retinopathy of prematurity	ND	observational design, lack of information on the daily type of feeding in the Mix group, indication for supplementation, and timing with the bovine and individual fortification of human milk or formula	Probiotic supplementation had protective and promotive roles on sepsis and growth, respectively
[114]	2020	Perth_ Australia	Retrospective cohort	Comparing clinical outcomes of *Bifidobacterium breve M-16V* supplementation in full-term and preterm infants	1380 (162 preterm, 1218 full term)	1 neonatal intensive care unit	preterm neonates (gestation <34 weeks) admitted between June 2012 and August 2015	ND	<34 weeks, subgroup: <29 weeks	<1500	*B. breve M-16* V (Morinaga Milk Industries, Tokyo, Japan)	3 × 10^9^ cfu/day	ND	gestational age, gender, duration of respiratory support, and patent ductus arteriosus were controlled as confounding factors	twice daily continued till the 37 weeks gestational age or discharge (Supplementation was stopped after suspected NEC or sepsis diagnosis)	NEC (stage ≥ 2), Los	No adverse effect by monitoring blood culture positive sepsis by B. breve M-16V, abdominal distension, vomiting, and diarrhea leading to the cessation of the supplementation	retrospective design that makes it difficult to determine the effects of known and unknown confounders, relatively small sample size	No significant difference was observed between the two groups regarding NEC rate, LOS, and mortality. postnatal age at full feeds was higher in preterm infants
[115]	2021	Iowa_ USA	single-center retrospective chart review	Evaluation of *Bifidobacterium* and *Lactobacillus* supplementation impact on NEC	37 (14 Pre-probiotic, 23 Post-probiotic)	1 Children’s Hospital	Probiotic group: Infants with at least 3 days old, born at <33 weeks gestational age, with a corrected post-menstrual age of at least 24 0/7 weeks who received intakeof at least 6mL of enteral feedings per day	Infants with major congenital abnormalities, anatomic obstruction of the gastrointestinal tract, inguinal hernia repair, G-tube placement, or peritoneal dialysiscatheter placement	<33 weeks	<1500 g	multispecies probiotic (Bifidobacterium breve, bifidum, infantis, and longum) plus Lactobacillus rhamnosus GG (Ultimate Flora Baby Probiotic)	2 × 10^9^ colony forming units per 0.5 g.	ND	randomly assigned (1:1) using a randomization program	Daily until the infant reaches a corrected post-menstrual age	NEC (Bell’s stage ≥2a)	NA	It was a retrospective study, which made it difficult to control confounders and interpret findings properly, single-center design	Probiotic supplementation did not have significant impact on the NEC rate and mortality
[116]	2021	USA	non-concurrent, retrospective cohort study	*Bifidobacterium infantis* EVC001 supplementation impact on NEC rate	483 (182 probiotics, 301 placeboes)	1 hospital	weight <1500g, received full resuscitation, and survived until day-of-life 3, (the earliest time at which very low birth weight infants received at least one feed of EVC001, fed human milk-based diets consisting of either mother’s milk, donor milk, or a combination, fed according to institutional guidelines incorporating best practices for NEC prevention, including a human milk-based diet, an initial period of trophic feeding and gradual feeding advancements, did not have hemodynamically significant congenital heart disease	underwent palliative delivery or unsuccessful resuscitation, died prior to day-of-life 4, were fed a non-human milk-based diet prior to 34 weeks PMA, immunodeficiency, received less than two feeds of EVC001 in EVC001 group	≅28 weeks	<1500	active B. infantis EVC001	8 billion colony forming units (CFU) in 0.5 mL of medium chain triglyceride oil	human milk-based diet of mother’s milk, donor milk, or both	adjusted for sex, birth weight, gestational age at birth, and mode of delivery	Daily prior to morning feed until 34 weeks post-menstrual age or for a minimum of two weeks, whichever duration was longer	NEC (≥Stage 2)	reported safe	observational design, absence of fecal sampling to confirm that *B. infantis EVC001* supplementation led to successful modulation of the preterm intestinal microbiota	Probiotic administration led to a significant reduction in the NEC rate and mortality
[117]	2022	Paris_ France	prospective longitudinal study	Characterization of *Bifidobacteria* strains isolated from preterm infants	26	1 pediatric hospital	Preterm infants with at least 2 fecal samples with bifidobacterial colonization at different times	Preterm infants with malformations or metabolic diseases	<37 weeks	From 710 to 2610 g	ND	ND	ND	ND	ND	ND	ND	Low genotype resolution	Environmental factors can affect phenotypes in *Bifidobacterium* strains. Phenotypes and genotypes of *Bifidobacteria* species were unstable during the first year of life. Twin infants have a more similar microbiota compared to other infants

ND not described.

## Data Availability

Not applicable.

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
