# Peer review of "Bifidobacterium: Host–Microbiome Interaction and Mechanism of Action in Preventing Common Gut-Microbiota-Associated Complications in Preterm Infants: A Narrative Review"

_nutrients, 2023, doi:10.3390/nu15030709_

Round 1

Reviewer 1 Report

This manuscript reviews the role of Bifidobacterium in the development of the gut microbiota of preterm infants and discusses the use of Bifidobacterium as a therapeutic for the treatment and prevention of common diseases in preterm infants. The manuscript is clearly written, and the subject is of interest and appropriate for the journal.

Minor comments

Lines 57 – 72: Scientific names should be in italics

Lines 104, 157, 384: when listing bacteria, the “and” should not be in italics

Line 251: remove “(“

Line 337: missing “)”

Line 365: use italics 5. Bifidobacterium as probiotic

Table 1: this table is difficult to read. Either improve the format or use it as a supplemental material

Line 494: the numbers for probiotic:40 and placebo:82 don’t match with the total 75 eligible infants mentioned. Review  

Line 496: ATCC 53103 should not be in italics

Line 558: use italics for “B.” of B. infantis and no for “and” for B. infantis and L. acidophilus

Conclusion: the conclusion is vague and general. The aim of this manuscript is to provide new insights for microbiome-targeted interventions for personalized medicine in preterm infants. The conclusion should be focused on that topic in addition to suggest future directions in the use of Bifidobacterium as a therapeutic for preterm infants.

References: use italics for scientific names in the references listed

Author Response

Dear reviewer

We really appreciate your time and helpful comments on our review paper. You made crucial points and responding to them greatly improved the manuscript. Please find point-by-point answers to your comments as follow:

Lines 57 – 72: Scientific names should be in italics

-Thank you for your comment, All scientific names are in italics which are highlighted in yellow in the manuscript (lines 60-63)

Lines 104, 157, 384: when listing bacteria, the “and” should not be in italics

 -Thank you for your attention, we have corrected this typo across the manuscript and highlighted it in yellow.

Line 251: remove “(“

 -It has been corrected and highlighted in lines 257-264

Line 337: missing “)”

 -It has been corrected and highlighted in line 353

Line 365: use italics 5. Bifidobacterium as probiotic

 -Bifidobacterium is corrected to italics in line 381

Table 1: this table is difficult to read. Either improve the format or use it as a supplemental material

 -Thank you very much for your comment, We will discuss this with the production team to format the table according to the journal requirement in a way that is easy to read for the readers since the table contains some important details discussed across the manuscript.

Line 494: the numbers for probiotic:40 and placebo:82 don’t match with the total 75 eligible infants mentioned. Review  

-Thank you very much for your attention, the correct sample size has been added in line 513 according to the original paper (probiotic:40 and placebo:35)

Line 496: ATCC 53103 should not be in italics

-Thanks for your attention it is corrected now in line 514.

Line 558: use italics for “B.” of B. infantis and no for “and” for B. infantis and L. acidophilus

-We have corrected this section highlighted in yellow in line 577

Conclusion: the conclusion is vague and general. The aim of this manuscript is to provide new insights for microbiome-targeted interventions for personalized medicine in preterm infants. The conclusion should be focused on that topic in addition to suggest future directions in the use of Bifidobacterium as a therapeutic for preterm infants.

-Thank you very much for your comment, We have modified the conclusion section as you suggested highlighted in yellow (lines 779 to 789).

References: use italics for scientific names in the references listed

-Thank you very much for raising this point, since we are using the Nutrients referencing style it automatically changes the names but we will discuss with the production team if we can use a different style for Endnote so it won't change the bacterial names in italics.

Reviewer 2 Report

Dear Authors,
Congratulations on submitting a very interesting and thorough narrative review on bifidobacterium and its role in the development of preterm microbiota.
Title
The title of the manuscript draws attention of the reader, defines the study population, and outcomes but is not clear on the type of manuscript. Please consider the following title:  Bifidobacterium: Host-microbiome interaction and mechanism of action in preventing common gut microbiota-associated complications in preterm infants. A narrative review.

Abstract
The authors provide a concise overview of the study. It is clearly stated why the study will be undertaken, and how the data will be collected.

Introduction
The authors provide enough contextual information for the journal’s nutritional and medical readership to understand the context. They accurately describe up to date research, with is properly referenced. Additionally, limitations and controversies are identified. 
Narrative review

Authors provide a thorough review of up to date literature, and relevant studies are adequately cited.

Discussion
The discussion addresses the research problem. Other relevant studies are discussed. 

Author Response

Dear reviewer

We really appreciate your time and helpful comment on our review paper.

You made a crucial point, and as you advised, we changed the title to more accurately reflect the type of the study to “Bifidobacterium: Host-microbiome interaction and mechanism of action in preventing common gut microbiota-associated complications in preterm infants. A narrative review”